# Nonlinear Inviscid Aerodynamics of a Wind Turbine Rotor in Surge, Sway, and Yaw Motions Using a Free Wake Panel Method

André F. P. Ribeiro[a], Damiano Casalino[a], and Carlos S. Ferreira[a]

[a]Delft University of Technology, Kluyverweg 1, Delft, Netherlands

**Correspondence:** A.F.P. Ribeiro (A.PintoRibeiro@tudelft.nl)

**Abstract.** We investigate the aerodynamics of a surging, heaving, and yawing wind turbine with numerical simulations based on a free wake panel method. We focus on the UNAFLOW case: a surging wind turbine which was modelled experimentally and with various numerical methods. Good agreement with experimental data is observed for amplitude and phase of the thrust with surge motion. We achieve numerical results of a wind turbine wake that accurately reproduce experimentally verified effects of surging motion. We then extend our simulations beyond the frequency range of the UNAFLOW experiments and reach results that do not follow a quasi-steady response for surge. Finally, simulations are done with the turbine in yaw and heave motion and the impact of the wake motion on the blade thrust is examined. Our work seeks to contribute a different method to the pool of results for the UNAFLOW case, while extending the analysis to conditions that have not been simulated before, and providing insights into nonlinear aerodynamic effects of wind turbine motion.

## Nomenclature

| | | | |
|---|---|---|---|
| BEM | blade element momentum theory | $U_S$ | rotor maximum surge velocity ($2\pi f A$) |
| CFD | computational fluid dynamics | $U_Y$ | maximum yaw tip velocity $2\pi f A R$ |
| FOWT | floating offshore wind turbine | $U_\infty$ | freestream velocity |
| $A$ | surge, sway, or yaw motion amplitude | $V_S$ | maximum sway velocity ($2\pi f A$) |
| $A_{ij}$ | doublets influence coefficients matrix | $V_{S,c}$ | sway velocity projected onto blade chord |
| $A_r$ | reduced amplitude ($A/D$) | $x$ | instantaneous rotor streamwise position |
| $B_{ij}$ | sources influence coefficients matrix | $\dot{x}$ | instantaneous rotor surge velocity |
| $C_{iw}$ | wake vortices influence coefficients matrix | $\beta$ | side wind angle |
| $C_T$ | rotor thrust coefficient | $\beta_{max}$ | maximum side wind angle due to sway motion ($\tan^{-1}(V_S/U_\infty)$) |
| $D$ | wind turbine rotor diameter | $\gamma$ | wake vortex strength |
| $f$ | surge, sway, or yaw motion frequency | $\Delta C_T$ | rotor thrust coefficient fluctuation amplitude |
| $f_r$ | reduced surge frequency ($f D/U_\infty$) | $\Delta C_{Tb}$ | blade thrust coefficient fluctuation amplitude |
| $f_\Omega$ | rotor rotation frequency | $\Delta \psi$ | rotor rotation angle in one timestep |
| $p$ | pressure | $\mu$ | doublet strength |
| $R$ | wind turbine rotor radius | $\rho$ | air density |
| $T$ | rotor thrust | $\sigma$ | source strength |

| | | | |
|---|---|---|---|
| $t$ | time | $\Phi$ | velocity potential |
| $U$ | surface flow velocity | $\phi$ | phase between rotor surge motion and thrust |
| $U_k$ | panel kinematic velocity | $\psi$ | blade azimuth angle |

## 1   Introduction

With the wind energy market leaning heavily towards offshore turbines in recent years, floating offshore wind turbines (FOWT) have become the focus of numerous research groups. One of the many challenges of such configurations is that, due to oceanic waves, the turbine is subjected to large amplitude motions, making its aerodynamics even more complex than that of onshore turbines. Turbines can translate horizontally perpendicular (surge) or parallel (sway) to the rotor plane. They can translate vertically (heave). They can rotate around the tower axis (yaw), or around the two horizontal axes (roll and pitch). These degrees of freedom are illustrated in Figure 1.

The sway and heave motion are, from a rotor aerodynamics perspective, equivalent. Rolling moves the rotor in a very similar way to sway, with an added in-plane rotation, equivalent to a change in rotation velocity. Pitching can be thought of as a combination of surge, yaw, and heave. Hence, for rotor aerodynamics, we can consider the surge, yaw, and sway as the fundamental forms of rotor motion, from which the others can be derived. For this reason, in this study, we focus on these three degrees of freedom. While these rotor motions have been studied experimentally (Fontanella et al., 2022), the frequencies and amplitudes of the motion are typically limited and inertial effects can affect the accuracy of the results. Hence, numerical studies are needed to investigate FOWT motion.

The UNAFLOW (Bayati et al., 2018b; Fontanella et al., 2021a) project provided a simplified test case for a non-stationary rotor, by simulating a surging wind turbine in a wind tunnel, without any tilting of the tower. Several groups have simulated the UNAFLOW case with different methodologies including blade element momentum theory (BEM), lifting line, and computational fluid dynamics (CFD), with fairly good results being achieved (Bayati et al., 2018a; Boorsma and Caboni, 2020; Cormier et al., 2018). Furthermore, vortex methods have shown promising results for FOWT in surge and other degrees of freedom for other turbines (Ramos-García et al., 2022b, a).

While BEM simulations have successfully captured dynamic inflow conditions (Mancini et al., 2023), most of the research has focused on dynamic blade pitch, streamwise velocity fluctuations, or surge. Recent developments have been made to extend BEM to general wind turbine motion (Mancini et al., 2022) but, to our knowledge, validation of these models for sway conditions have not been extensive. Dynamic yaw and sway motion have the potential to be more difficult to capture than surge, as the wake moves from side to side and the assumptions of momentum theory may lead to large errors.

This work is an expansion of what was documented a previous conference publication (Ribeiro et al., 2022b). For a full description of the numerical methods, along with verification and validation on relevant cases, refer to that paper. Here, we seek to contribute to the pool of UNAFLOW results by simulating the UNAFLOW case with a source and doublet free wake panel method. Unlike BEM and lifting line, panel methods directly model the blades, free from table look ups, while still being a fraction of the cost of a CFD simulation (Leishman, 2002). Blade thickness effects are included, by simulating the entire

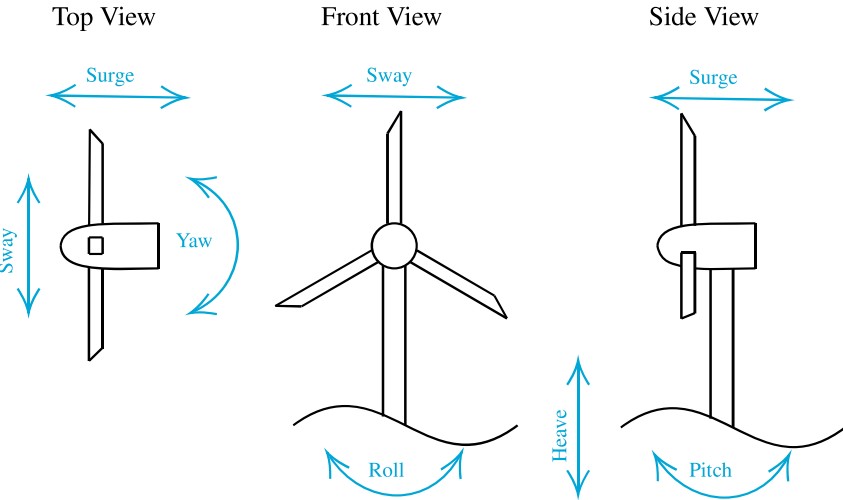

**Figure 1.** Degrees of freedom of a FOWT.

blade surface, rather than the camber surface or a single line, which can lead to better accuracy (Yang et al., 2020). Panel methods have also been shown to accurately model full rotors, including aeroelastic effects (Gennaretti et al., 2018; Sessarego et al., 2015; Wang et al., 2016). The free wake allows for complex scenarios, such as blade vortex interaction (Gennaretti and Bernardini, 2007), which could happen in extreme surge conditions.

With these characteristics in mind, this is an important stepping stone towards the ultimate goal of this research: aeroelastic simulations of FOWT through a fully-coupled transient aerodynamic/structural fluid-structure interaction (FSI). To our knowledge, only experimental and CFD results have been used to investigate the wake of the UNAFLOW turbine (Bayati et al., 2018a). CFD adds significant diffusion to the tip vortices, making comparisons to experiments difficult. Hence, in this work we also show how the free wake panel method compares to experimental measurements of the wake.

The next objective of this work is to extend the surge analysis to sway and yaw motion. We use the UNAFLOW rotor to perform such investigations, in order to contribute to the knowledge of the physics of these motions. Finally, we seek to understand the impact of the wake motion on surge, sway, and yaw. We do this by employing unique features of the free wake panel method, allowing us to include rotor motion effects indirectly. The analysis of wake motion effects seeks to clarify mechanisms of turbine motion that will need to be accounted for when simulating FOWT motion with methods without true
wake motion, such as BEM and prescribed wake vortex methods.

## 2 Methodology

We employ a source and doublet panel method with free-wakes (Katz and Plotkin, 2001) in order to capture the aerodynamics of the surging wind turbine rotor. The thickness effects are fully captured, as the panels lie on the blades surfaces. Blade-vortex interaction capabilities (Gennaretti and Bernardini, 2007) are implemented, but not used for this study, as the turbine does not cross its own wake. The surface and wake discretization of the velocity potential equation leads to the following linear system:

$$\frac{1}{4\pi}A_{ij}\mu_j + \frac{1}{4\pi}B_{ij}\sigma_j + \frac{1}{4\pi}C_{iw}\gamma_w = 0 \tag{1}$$

where $A$, $B$, and $C$ are the influence coefficients matrices (Maskew, 1987) for the doublets $\mu$, sources $\sigma$, and wake vortices $\gamma$ respectively. The values of $\mu$, $\sigma$, and $\gamma$ are constant over each panel. The sources $\sigma$ are computed to ensure impermeability, the wake vortices $\gamma$ enforce the Kutta condition (Youngren et al., 1983), and the doublets $\mu$ are the unknowns. At every timestep, wake vortices are convected due to the freestream velocity and the induction of all the surface and wake panels.

When symmetries are present, as in turbines with multiple blades and no yaw or heave, virtual bodies across symmetry planes or axes can be used (Katz and Plotkin, 2001), which dramatically reduce the influence coefficients matrices. With the linear system solved, surface velocities $U$ are computed based on the basic potential flow equation, $U = -\nabla\Phi$, where $\Phi$ is the velocity potential. The surface gradient is computed with central differences for quadrangular panels, but a least squares approximation (Anderson and Bonhaus, 1994; Sozer et al., 2014) is also available, and is always used for triangular panels. With the surface velocity available, the unsteady Bernoulli equation (Bernardini et al., 2013) is used to find the surface pressure, which is then integrated over all surface panels to find the forces and moments acting on the bodies. The time derivative in the unsteady Bernoulli equation is calculated with a first order backwards Euler method.

The panel method shown here was created with the intent to be faster than available methods by use of efficient algorithms and modern acceleration techniques, such as leveraging GPUs for the calculations. At the moment, our implementation was compared to an open source C++ panel code (Baayen, 2012) and we observed over an order of magnitude in speed up. The method also supports large displacement aeroelasticity (Ribeiro et al., 2022a), although this is not included in this work.

## 3 Surging Wind Turbine Simulations

The UNAFLOW turbine case (Fontanella et al., 2021a) consists of a 3 blade rotor with a diameter of 2.38 m, rotating at 150 to 265 RPM, with $U_\infty$ between 2.5 and 6 m/s. The entire rotor surges upwind and downwind at a frequency ($f$) ranging from 0.125 to 2 Hz and amplitude ($A$) from 2.5 to 125 mm, with the rotor center axial position following $A\sin(2\pi f t)$, where $t$ is time. In non dimensional terms, for the case with $U_\infty = 4$ m/s, this corresponds to reduced frequencies $f_r = fD/U_\infty$ between 0.07 and 1.2 and normalized amplitude $A_r = A/D$ between 0.001 and 0.05. This motion is performed such that the rotation axis is always aligned with the freestream, meaning no yawed flow occurs. As the majority of the experimental data are for $U_\infty = 4$ m/s and RPM of 241, these are the conditions we simulate in this work. We use different values of $f$ and $A$ for our

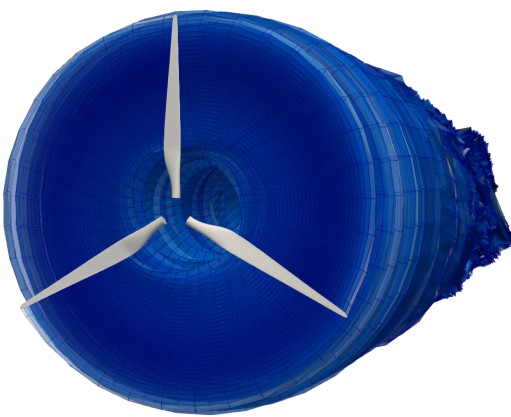

**Figure 2.** Panel method simulation results of the UNAFLOW rotor and its wake.

simulations, based on availability of experimental data, while giving a preference to cases with high surge velocity $U_S = 2\pi f A$. For details on full scale values for these quantities see Ribeiro et al. (2022b).

The blades are based on the SD7032 airfoil section, transitioning into a circle in the root region. For the simulations in this work, the blade geometry was constructed based on chord and twist distributions provided in the experimental data set (Fontanella et al., 2021b). However, this led to small differences in the geometry. In particular, the blade chord approaches zero at the tip (Bayati et al., 2016), which is inconsistent with the geometry description (Fontanella et al., 2021b), and can lead to some differences in results. Figure 2 shows the UNAFLOW wind turbine, along with its wake, as simulated by the methods described in this paper. The blades are simulated without the hub and tower for simplicity.

The blades are discretized with 100 chordwise panels, using a cosine distribution, and 50 equally spaced spanwise panels each, with a total of about $15,000$ panels. The panel distribution is shown in Figure 3. This panel distribution was chosen as it provided grid converged results for several preliminary studies on airfoils and rotors, not included here for brevity. The blade tips and roots are closed to enforce impermeability. Wake panels are only shed from the trailing edges, meaning no vorticity can be shed from the $90°$ edges at the tips and roots. The timestep is set to $1/36$ of a revolution unless otherwise stated, which corresponds to a rotation angle of $\Delta\psi = 10°$. Simulations are run for at least 40 revolutions, leading to about $216,000$ wake panels. This time discretization corresponds to 72 timesteps per surging period for $f = 2$ Hz, which is the highest surge frequency available in the experimental data set.

For a demonstration of the accuracy of the chosen timesteps, simulation duration, and validation of the mean flow properties, refer to Ribeiro et al. (2022b). Here, we focus on the main results for surge, while normalizing the plots in a different way than in the original publication, as this will help with comparisons to the other rotor motions. The main parameter we will use throughout this work is the thrust coefficient $C_T$:

$$C_T = \frac{T}{\frac{1}{2}\rho\pi R^2 U_\infty^2} \tag{2}$$

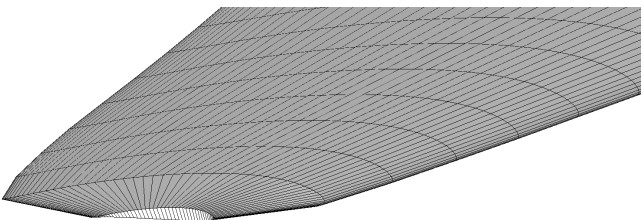

**Figure 3.** Surface mesh used for the UNAFLOW blade, showing tip region.

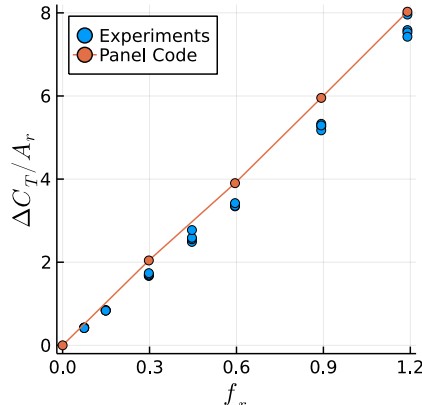

**Figure 4.** Surge frequency effect on the amplitude of the fluctuation of the thrust coefficient. Simulations at constant $A$, experiments at various $A$ shown.

**Figure 5.** Surge frequency effect on the phase between the rotor position and its thrust. Simulations at constant $A$, experiments at various $A$ shown.

where $T$ is the thrust force, $\rho$ is the air density, and $R$ is the turbine radius.

## 3.1 Surge velocity effects

We examine the fluctuating component of $C_T$, while keeping $A=15$ mm ($A_r=0.006$), and varying $f$ between 0 and 2 Hz ($f_r$ between 0 and 1.2). Figure 4 shows the effect of $f_r$ on the amplitude of the fluctuations of thrust ($\Delta C_T$), normalized by $A_r$. Note that $\Delta C_T$ is the amplitude that would multiply a sine function to represent the time history of $C_T$, or $(C_{Tmax}-C_{Tmin})/2$, assuming a time history of $C_T$ that is perfectly sinusoidal. Figure 5 shows the effect of $f_r$ on the phase shift $\phi$ between the rotor position and the fluctuations of $C_T$. The experimental data shown throughout this paper were filtered at the surge frequency. The lowest surge frequency, $f_r=0.3$ Hz was run for twice as long as other cases, to obtain meaningful statistics from the simulations.

The values of $\Delta C_T/A_r$ agree well with experimental data, with an approximately linear relation between the surge velocity and the thrust fluctuations. This confirms the quasi-steady nature of the results, which is likely due to the relatively small values of $U_S$ (Mancini et al., 2020). The average slope of $\Delta C_T/A_r$ as a function of $f_r$ in the simulations is 15% higher than in the

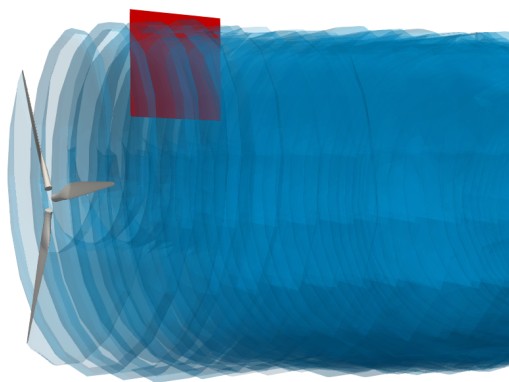

**Figure 6.** UNAFLOW wind turbine (grey), wake panels (blue), and PIV plane (red).

experiments, likely due to the inviscid approach. The values of $\phi$ fall within the experimental scatter, being within $3°$ of $-90°$ for all cases, which corresponds to the quasi-steady response.

## 3.2 Rotor wake

We now focus on the rotor wake. The UNAFLOW experiments included particle image velocimetry (PIV) on a vertical plane in the rotor wake, aligned with the center of the nacelle. Measurements were made at several stages of the surging motion and averaged over several snapshots, with the rotor always being in the same azimuth (Fontanella et al., 2021a). We focus on two rotor positions, which the experiments refer to as steps 1 and 5. Both steps correspond to the rotor being in its central position ($x=0$), with the rotor moving with maximum velocity against the wind in step 1 ($\dot{x}=-U_S$) and maximum velocity with the wind in step 5 ($\dot{x}=U_S$). As the rotor is in the same position and same azimuth for both steps, any change in the wake is caused by unsteady effects of the surging motion.

Simulations are done with $f=1$ Hz and $A=35$ mm, which correspond to $f_r=0.6$, $A_r=0.015$, and $U_S/U_\infty=0.055$. This configuration was selected since it contains PIV data for all steps, while also having a high value of $U_S$. Figure 6 shows the UNAFLOW rotor, along with the wake panels and the PIV plane, as a reference for the results discussed in the following paragraphs. When the rotor crosses steps 1 and 5, the bottom blade is rotated $187°$ away from the PIV plane.

Figures 7 and 8 show experimental and numerical results on the PIV plane. Both steps 1 and 5 are shown in each figure, in order to better see the difference between them. The precise location of the tip vortices in the simulation is highlighted as pink points in both figures, for comparisons. Consistent with expectations, during step 1 the tip vortices are further downstream than in step 5. The horizontal distance between the vortices in steps 1 and 5 is very well captured by the simulation, being about 6 cm in both simulations and experiments. The vortices radial and streamwise positions are noticeably different between simulations and experiments, likely in large part due to the blades tip geometry not being identical to the experimental blades, as they are not described in details in the documentation. The blades in the simulations are slightly longer and have a larger

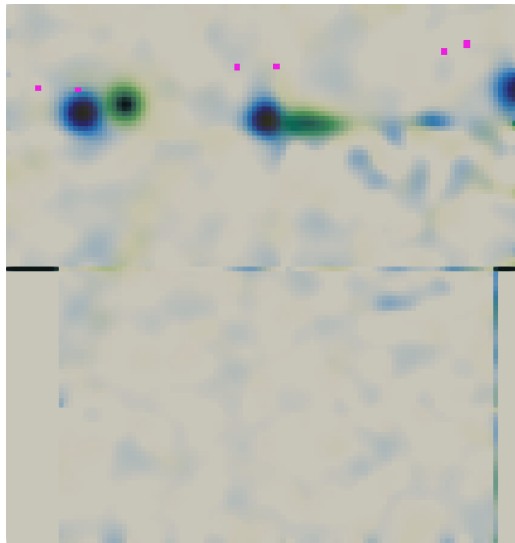

**Figure 7.** Experimental tip vortices position on steps 1 (green) and 5 (blue). Vorticity perpendicular to the plane shown from 0 to 300 1/s. Pink dots represent the tip vortices from the panel method simulation.

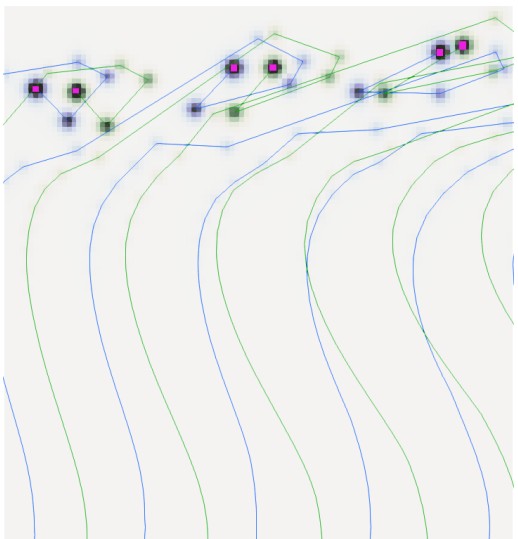

**Figure 8.** Numerical wake on steps 1 (green) and 5 (blue). Vorticity perpendicular to the plane shown from 0 to 300 1/s. Lines represent a cut through the wake panels. Pink dots highlight the tip vortices.

chord on the tip. It should be noted that the PIV data indicate the wake is shrinking in radius, which should not be the case and is likely an effect of experimental uncertainty. The expanding wake in the simulations is more realistic.

CFD simulations conducted for the UNAFLOW turbine (Bayati et al., 2018a) were able to capture the horizontal displacement of the first tip vortex in Figures 7 and 8 to some extent. However, the second tip vortex displacement was inverted, that is, the vortex from step 1 was upstream of the position in step 5. To our knowledge, no other studies have been made where the wake of a surging wind turbine was simulated numerically and the results were validated with experiments. Hence, we believe this is the first time that a surging wind turbine simulation shows results that agree well with experimental data in terms of wake dynamics.

It is worth noting that the wind turbine wake is folding upon itself on the right side of Figure 8. This is a common consequence of using an inviscid free wake method, as complex wakes tend to become tangled as they develop, which can be partially observed on the right side of Figure 2. Using a vortex core model (Ramasamy and Leishman, 2007) can stabilize the wake for a longer time if the vortex core radius is large enough, but has small effects on the location of the tip vortices in the plane investigated here. The simulations shown here employed the aforementioned vortex core model and achieved better wake stability with it. Without a vortex core model, the wakes became entangled at earlier locations.

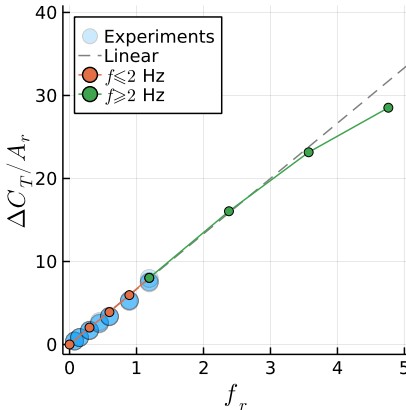

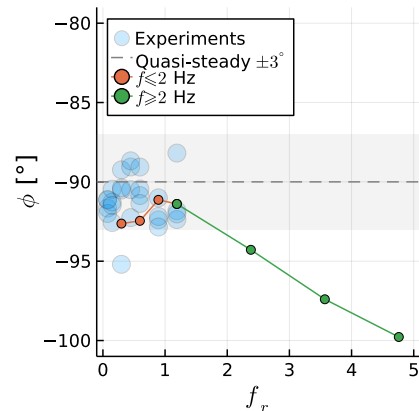

**Figure 9.** Maximum surge velocity effect on the amplitude of the fluctuation of the thrust coefficient.

**Figure 10.** Maximum surge velocity effect on the phase between the rotor position and its thrust.

### 3.3 Beyond the UNAFLOW Results

In this Section, we seek to expand our simulations beyond the limitations of the UNAFLOW experiments. Due to the relatively small reduced frequencies involved in wind turbine surge motion (Tran and Kim, 2016) and the assumptions of inviscid flow in the simulations, we do not expect superlinear increases in $\Delta C_T$. Based on Theodorsen Theodorsen (1934) results on airfoil sections, we expect first to see nonlinear behavior in the phase, and then a sublinear change in $\Delta C_T$. For more details, see Ribeiro et al. (2022b).

Results for simulations beyond the experimental data are shown in Figures 9 and 10. The orange circles represent the data shown in Section 3.1, with $A = 0.015$ mm and $f$ varying between $0.5$ and $2$ Hz. The green circles show an extension of the orange data, with the same $A$, but with $f$ varying from 2 to 8 Hz. The grey dashed line in Figure 9 is a linear extension of the orange circles. The grey region in Figure 10 represents the quasi-steady regime, which we set around $-90° \pm 3°$, as this contains most of the experimental data.

We can now find where $\Delta C_T / A_r$ breaks from a linear trend and when $\phi$ leaves the quasi-steady regime. At $f_r = 3.6$, we observe a noticeable deviation from the linear relation between $\Delta C_T / A_r$ and $f_r$. Increasing $f_r$ leads to a reduction in $\Delta C_T$, instead of the increase seen in 2D. In contrast $\phi$ moves away from the quasi-steady regime at an earlier point, near $f_r = 2$.

  Normalizing $\Delta C_T$ with the surge amplitude and plotting results against the frequency is sufficient to collapse the experimental data here, as demonstrated in previous studies. For a discussion on how to normalize surging rotor data, see (Mancini
et al., 2020). However, the normalization used herein for $f_r$ might not be sufficient to collapse nonlinear data with different flow conditions. If the nonlinear effects occur due to Theodorsen effects as is the case here, the rotor RPM should likely be taken into account, as it is an important factor for the value of $k$ along the blades.

## 4 Swaying and Yawing Wind Turbine Simulations

We now move on to simulations of the two other degrees of freedom of interest for FOWT in this work: sway and yaw. We
continue to use the UNAFLOW turbine, in spite of no experimental data being available for the cases investigated in this
Section. Although not shown, some of the simulations in Section 3 used symmetry conditions, with only a single blade of the
rotor being simulated. This is no longer possible, as the introduction of lateral wind makes the loads on the blades asymmetric.
Hence, all simulations in this Section include all three blades.

### 4.1 Fixed Turbine with Side Wind

In order to identify the dynamic effects of sway and yaw, we first need to understand the static effects of side wind. Hence,
we simulate a fixed UNAFLOW rotor with side wind. This is usualy referred to as a yaw case, but to avoid confusion between
static and dynamic yaw cases, we refer to the static yaw cases as side wind throughout this paper and use the word yaw to refer
to dynamic rotation around the tower axis.

We perform side wind simulations by rotating the wind vector around the vertical axis by a side slip angle $\beta$ varying
between 0 and $40°$, see Figure 11. Even though this is a static simulation, results can not converge to a steady state, as the
blades experience different wind vectors during a rotation, making cases with side slip intrinsically unsteady. However, as
there are three blades, the dynamic loads on the blades mostly cancel each other out, leading to negligible fluctuations of rotor
thrust. This occurred in our simulations for side wind, sway and yaw. Hence, instead of investigating $C_T$ on the entire rotor,
we instead look at it on a single blade, or $C_{Tb}$. This value becomes periodic as the simulation progresses and its fluctuation
amplitude $\Delta C_{Tb}$ will be investigated in this and the following Sections.

Figure 12 shows the effect of $\beta$ on $\Delta C_{Tb}$. We see a linear trend, with the thrust fluctuations increasing as $\beta$ increases up
to about $10°$. After that, the results are nonlinear and, as in surge, fall below the linear trend. For the high values of $\beta$ we
investigate here, flow separations would likely occur in real turbines, increasing $C_{Tb}$. We highlight that for side wind, $C_{Tb}$
varies periodically as a sine wave, with the frequency of the rotor rotation. This serves as a baseline for the results that follow.

### 4.2 Swaying Turbine

We impose a swaying motion on the turbine using the same conventions of Section 3. We set $A_r = 0.05$ and conduct simulations
at $f$ between 1 and 5 Hz. Again we use $\Delta C_{Tb}$ to measure the sensitivity of the blades to the unsteady loads. In this and the
next Sections, we start the rotor motion with one blade pointing straight up, and refer to this as "blade 1". The other blades are
numbered in clockwise direction, looking downwind. The choice of the blade can affect the results, as will be shown.

The sway motion introduces a side velocity, which at its maximum value (when the rotor is at the center of the motion)
$V_S = 2\pi f A$, introduces a maximum side wind angle $\beta_{max} = \tan^{-1}(V_S/U_\infty)$, see Figure 13. We can use $\beta_{max}$ to compare the
sway results to the static results of the previous Section. Note that the sway velocity adds to $U_\infty$, meaning $C_{Tb}$ must be scaled
with a higher incoming velocity, or multiplied by $cos(\beta_{max})^2$ for a fair comparison with the side wind case, which we do.

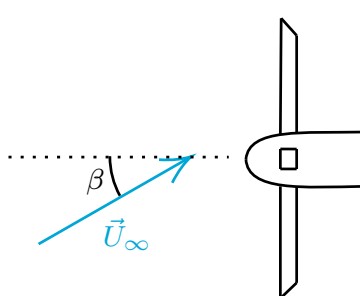

**Figure 11.** Top view of turbine with definition of side slip angle $\beta$. Dotted line represents the rotor axis.

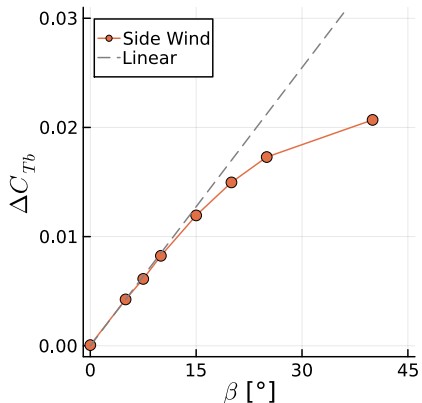

**Figure 12.** Axial force fluctuation amplitude of single blade during rotation for different side wind angles.

Figure 14 shows the time history of $C_{Tb}$ for blade 1 at different values of $f$. The signals repeat periodically and we show one period for clarity. The mean values of $C_{Tb}$ reduce with frequency due to the scaling explained in the previous paragraph. The forces on the blades actually increase with $f$. We see that, unlike the surge and side wind results, more than one frequency is involved in the response to sway motion. Figure 15 shows the amplitude of each sway frequency, as a function of the maximum $\beta$ experienced during sway, along with the side wind results of the previous Section. The effects become nonlinear for values of $\beta_{max}$ similar as seen in the side wind results. A substantial discontinuity appears for $\beta_{max} = 40°$, where $f$ is equal to the rotation frequency, making the response in Figure 14 a simple sine wave and reducing $\Delta C_{Tb}$.

By scaling $\Delta C_{Tb}$ with $cos(\beta_{max})^2$, we are able to match the linear regions of the sway cases with the side wind results. Note that without this scaling, $\Delta C_{Tb}$ grows in a superlinear fashion, instead of the sublinear trend seen in Figure 15. Different studies may or may not include such a factor, which would potentially lead to conflicting conclusions.

We now take a closer look at one of the curves of Figure 14, to explain their behavior. We take the case at $f = 2$ Hz because it is relatively simple, due to its sway frequency being half of the rotation frequency, but complex enough that it can serve as an example to explain all the other curves. Figure 16 shows a zoomed in view of $C_{Tb}$ as a function of time, where the blade azimuth is shown as vertical lines. Figure 17 shows the horizontal rotor sway velocity projected onto the blade chord $V_{S,c}$ in the same intervals and with the same vertical lines. The color conventions for the azimuth $\psi$ are shown in Figure 18.

The horizontal rotor sway velocity varies with the cosine of the sway frequency $f$, while the projection onto the chord varies with the cosine of the rotational frequency $f_\Omega$. When both functions are at their maxima with opposite signs, the maximum blade thrust is achieved, as seen at $9.75$ s. When they are at their maxima with the same sign, the blade thrust is minimized, as seen at $9.5$ s. The equation for the horizontal sway velocity projected onto the blade chord is:

$$V_{S,c} = -2\pi f A \cos(2\pi f t) \cos(2\pi f_\Omega t) \tag{3}$$

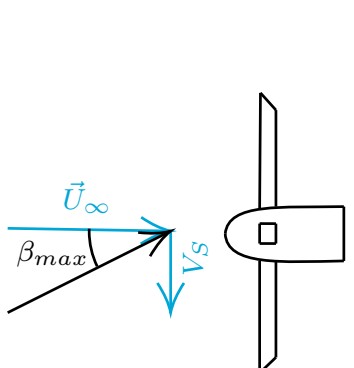

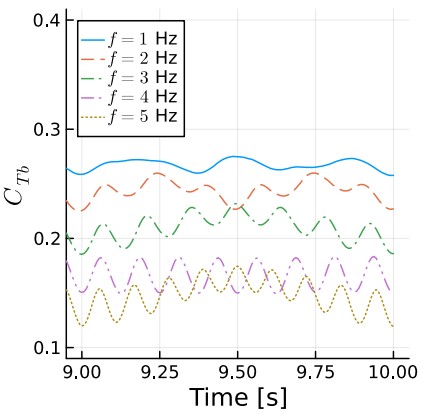

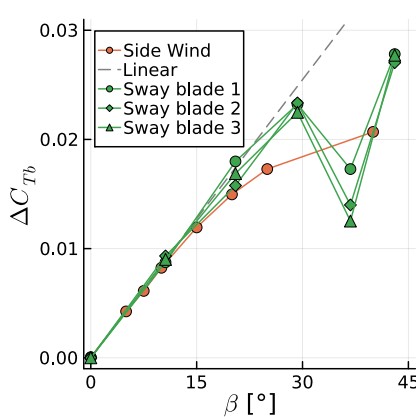

**Figure 13.** Top view of turbine with definition of maximum side wind angle $\beta_{max}$.

**Figure 14.** Time history of blade 1 thrust coefficient for various sway frequencies. Thrust normalized with maximum rotor velocity magnitude.

**Figure 15.** Amplitude of blade thrust coefficient fluctuation for various side wind angles and various maximum side wind angles achieved during sway motion. Thrust normalized with maximum rotor velocity magnitude.

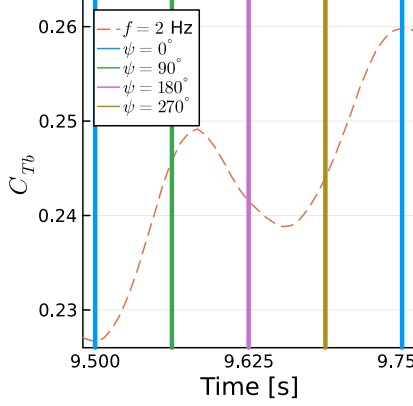

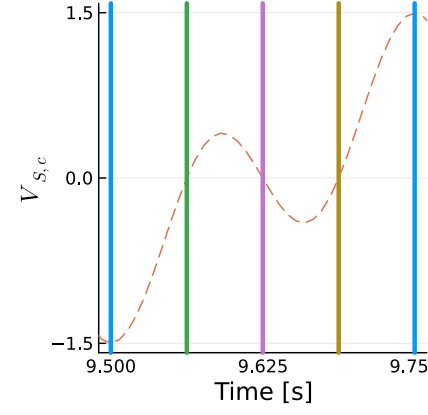

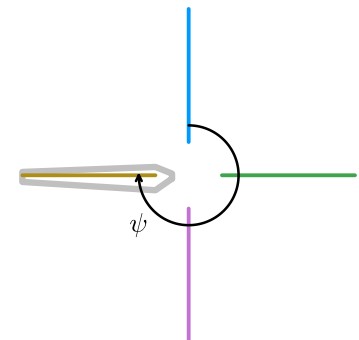

**Figure 16.** Time history of blade 1 thrust coefficient for a sway case. Vertical lines show blade azimuth.

**Figure 17.** Time history of horizontal sway velocity projected on airfoil chord at the tip of blade 1. Vertical lines show blade azimuth.

**Figure 18.** Azimuth convention used. Blade shown at $\psi = 270°$.

We can see that the thrust fluctuations in Figure 16 are nearly directly proportional to $V_{S,c}$, as seen in Figure 17, other than some differences that we will address in Section 4.4. All the other curves in Figure 14 can be interpreted in similar ways. They are all at $\psi = 0°$ at multiples of $0.25$ s.

The general behavior for sway motion is as follows: when the blade is pointing to either side, the sway velocity projection onto the blade chord is zero, sway effects are minimal, and the thrust is near its mean value. When the blade is pointing up or down, the potential for sway effects is maximum and the thrust can reach its maximum or minimum, if this coincides with maximum sway velocity. Hence, the relationship between the rotation and sway frequencies, along with the phase between the trigonometric functions that represent those motions will dictate the behavior of the blade thrust.

## 4.3 Yawing Turbine

We impose a yawing motion on the turbine by rotating it around the vertical central axis using the same conventions of Section 3. This results in a dynamic angle between the freestream velocity and the rotor axis, as in Figure 11. The yaw amplitude is the maximum value of this angle, which we set to $A = 3°$. Once again, the simulations are conducted $f$ varying between 1 and 5 Hz. The yawing motion introduces a velocity on the rotor relative to the rotation axis, which reaches its maximum at the tip radius $R$:

$$U_Y = 2\pi f A R \tag{4}$$

where $A$ must be in radians. As $U_Y$ can act with or against $U_\infty$, we do not adjust the flow velocity in the calculation of $C_{Tb}$.

Figure 19 shows the time history of $C_{Tb}$ for different values of $f$. Results are remarkably similar to Figure 14, with multiple frequencies involved and the pure sine response for $f = 4$ Hz. Figure 20 shows $\Delta C_{Tb}$ as a function of $U_Y/U_\infty$ for the yaw cases. Results seem to indicate a linear trend throughout all cases, with some effects of blade choice and again a large decrease in thrust fluctuation when the yaw and rotation frequencies match, at $U_Y/U_\infty = 0.4$, as in sway. Surge cases are also included for reference, as a function of $U_S$. This is the main reason for using $U_Y$ in Figure 20, as it allows us to compare surge and yaw in the same graph. If we used the reduced frequency, this would not account for the different motion amplitudes used in either case.

Note that $U_Y$ is not acting on the entire rotor plane. It acts mostly near the blade tips when they are horizontal. Hence we see that in spite of the more complex nonlinear behavior of the yaw motion, for comparable $U_S$ and $U_Y$, the surging motion is more critical for blade loading. Taking into account phase cancellation effects of the three blades for yawing motion, the rotor forces (but not necessarily the rotor moments) acting on the tower will be less critical in yaw versus a comparable surge motion as well. Also noteworthy is that yaw seems to remain linear even at $U_Y/U_\infty = 0.5$, whereas the surge becomes nonlinear around $U_S/U_\infty = 0.15$.

Similar to the previous Section, we now focus on a single yaw case, namely $f = 3$ Hz, and zoom into the time history of $C_{Tb}$. This is shown in Figure 21. The same conventions of Figure 18 are used. Figure 22 shows the streamwise component of the blade tip yaw velocity $U_{Y,R}$. The forces fluctuations are nearly proportional to $-U_{Y,R}$, as they are modulated by the yaw frequency $f$ and rotational frequency $f_\Omega$. Again, as in sway, the thrust fluctuation is nearly perfectly proportional to the velocity induced by the motion, although inversely proportional in this case. The blade tip velocity introduced by yaw as a function of time $t$ is:

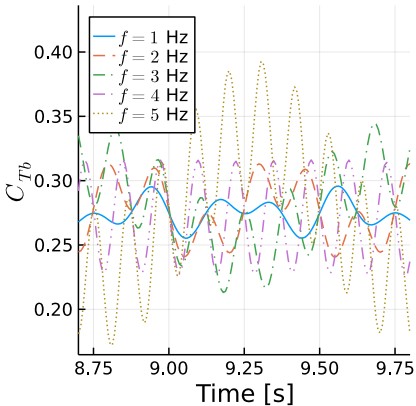

**Figure 19.** Time history of blade 1 thrust coefficient for various yaw frequencies.

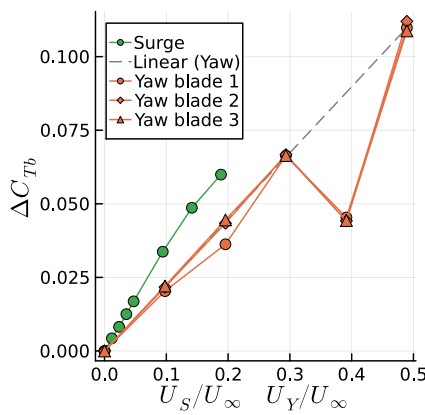

**Figure 20.** Amplitude of blade thrust coefficient fluctuation for various maximum surge velocities $U_S$ and maximum tip yaw velocities $U_Y$.

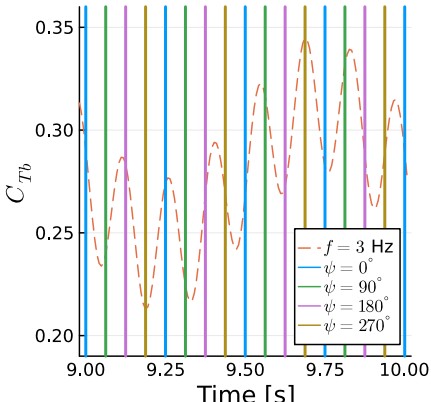

**Figure 21.** Time history of blade 1 thrust coefficient for a yaw case. Vertical lines show blade azimuth.

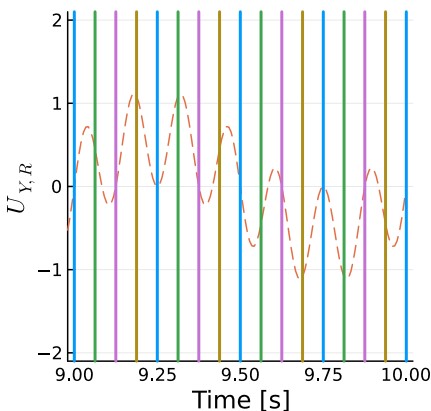

**Figure 22.** Time history blade 1 streamwise blade tip yaw velocity. Vertical lines show blade azimuth.

$$U_{Y,R} = 2\pi f A \cos(2\pi f t) R \sin(2\pi f_\Omega t) \tag{5}$$

where the cosine corresponds to the velocity change due to yaw and the sine is due to the blade tip distance to the yaw axis changing with the blade azimuth. A negative $U_{Y,R}$ is in the opposite direction as the freestream velocity, adding thrust.

The general behavior for yaw is as follows: when the blade is pointing up or down, the velocity introduced by the yaw force is zero and the thrust is near its mean value. This can be seen for all cases of $\psi = 0°$ and $\psi = 180°$. When the blade is at $\psi = 90°$ or $\psi = 270°$, there is a potential for high yaw effects, if this coincides with high yaw velocities. This can be seen near 9.2 and 9.3 s. In other words, the amplitude of the thrust fluctuations is linked to the phase between the trigonometric functions describing the blade rotation and the yaw motion.

## 4.4  Effect of Blade Azimuth

The results shown so far for sway and yaw are all for single blades, which had specific initial position, with blade 1 pointing up. The sway and yaw motions were all done with the rotor in neutral position at time equal to zero. The rotation frequency was 4 Hz and the sway and yaw frequencies were 1, 2, 3, 4, and 5 Hz. Hence, the rotor motion and blade position were locked in phase and this is not representative of all the possible loads blades can experience with different starting positions or with non-integer frequency ratios.

We can analyse Equations 3 and 5 to verify the amplitude of $\Delta C_{Tb}$ for arbitrary combinations of frequencies and phase. We can see that $V_{S,c}$ is a function of the product of two cosines and $U_{Y,R}$ a product of a sine and a cosine. Let us refer to these products as the phase-frequency amplitude, as they are linked to the phase of the trigonometric functions and the ratio of their frequencies. In both cases, the trigonometric products can vary in amplitude between $0.5$ and $1$, where $1$ corresponds to the amplitude varying between $-1$ and $1$. Hence, we can use the previous results, calculate the phase-frequency amplitude for each case, normalize the results by the phase-frequency amplitude, and then multiply them by $0.5$ and $1$. This will allow us to see the range of possible results for each sway and yaw amplitudes, for arbitrary combinations of frequencies and phase.

Figures 23 and 24 show the numerical results achieved for each blade in the previous simulations, along with the theoretical range of the results for arbitrary combinations of frequency ratios, and starting blade position. The dashed lines represent the possible range of results based on each blade result with corresponding colors. For the yaw motion, in Figure 24, we see that the range calculated based on each of the blades is always nearly identical. Hence, the thrust amplitude is behaving according to Equation 5. The same can be said for most of the sway motion in Figure 23, however at $\beta_{max} = 37°$, i.e. when the sway frequency is identical to the rotation frequency, all three curves are supposed to be at the bottom range of the prediction of Equation 3, but in reality they are at different levels. Hence, there is a mismatch between the three ranges, showing that the sway results are not simply following Equation 3, which is hinted at by the fact that the trends are nonlinear.

The thrust fluctuations start by scaling linearly with the surge, sway, and yaw motions, as these motions act on the blade sections, increasing and decreasing the relative flow velocity. The surge and yaw motions act on the axial velocity of a given blade section, while the sway motion changes the tangential velocity on the blade sections. As the tangential velocity is typically much higher than the axial velocity in wind turbines, the sway motion effects are small, compared to the surge and yaw. However, the sway motion moves the rotor out of the slipstream, leading to changes in axial velocity that are not due to simple changes in the kinematic velocities. This effect is quantified in the next Section.

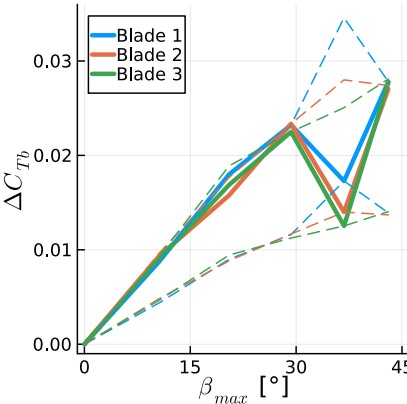

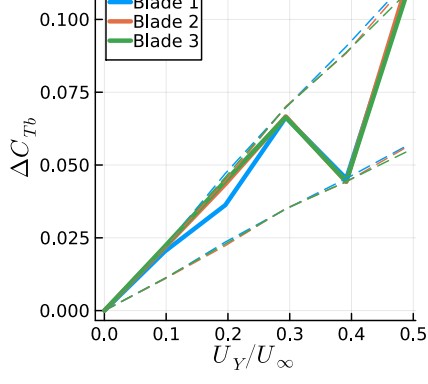

**Figure 23.** Amplitude of blade thrust coefficient fluctuation for various maximum side wind angles achieved during sway motion. Simulations results shown, along with the theoretical possible range of results in dashed lines. Thrust normalized with maximum rotor velocity magnitude.

**Figure 24.** Amplitude of blade thrust coefficient fluctuation for various maximum tip yaw velocities. Simulations results shown, along with the theoretical possible range of results in dashed lines.

## 5 Wake Motion Sensitivity Study

Methods such as BEM are unable to capture the detailed wake motion of wind turbines shown in Figure 7, while the numerical dissipation of CFD also introduces challenges for preserving tip vortices. However, such methods are able to achieve accurate results for rotor motion (Mancini et al., 2020). Hence, we seek to quantify the impact of the wake induction on the blades for moving rotors. We achieve this by comparing the simulations in the previous sections with cases where the rotor is not surging, swaying, or yawing, but the effects of these motions are still present. Throughout this section we refer to these simulations as having pseudo motion.

To do this, we take advantage of the properties of panel methods and model the rotor motion indirectly. The rotation of the rotor is still performed explicitly, but the surge, sway, and yaw motion are included not by displacing the turbine, but by modifying the equation for the sources $\sigma$ and the unsteady Bernoulli equation, which computes the pressure $p$. In both equations, the panel kinematic velocity $\boldsymbol{U}_k$ is used. Hence, for the simulations with real motion in the previous sections, the rotors were displaced and their displacement was then included in $\sigma$ and $p$ as $\boldsymbol{U}_k$. For the pseudo motion simulations, we add the surge, sway, and yaw velocities to $\boldsymbol{U}_k$ for computing $\sigma$ and $p$, while not surging, swaying, and yawing the rotor.

The pseudo motion method means that the wake panels are always released from the trailing edges in the fixed rotor position. The wakes are not identical to the wakes of a fixed rotating turbine, as changes in the circulation on the blades will affect how the wake is convected. However, the wakes are substantially different from the cases with real motion, while still being more realistic than a frozen or prescribed wake method. With this, we seek to quantify the effect of real motion and the associated realistic wake, compared to pseudo motion and the more simple wake that comes with it.

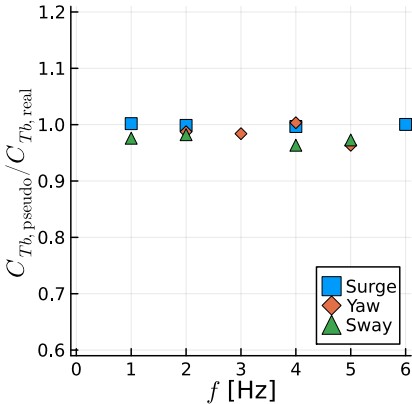
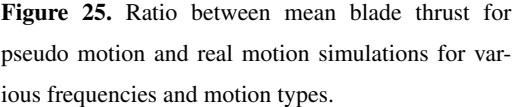
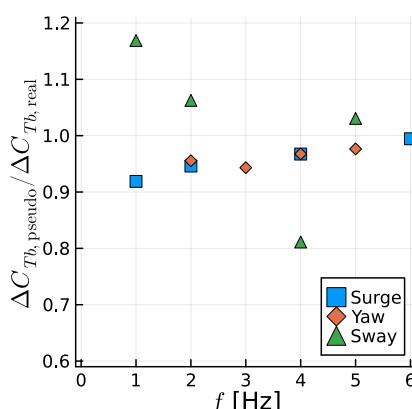

**Figure 25.** Ratio between mean blade thrust for pseudo motion and real motion simulations for various frequencies and motion types.

**Figure 26.** Ratio between amplitude of fluctuating blade thrust for pseudo motion and real motion simulations for various frequencies and motion types.

We select various frequencies from the previous Sections and simulate them in pseudo motion. The results are summarized in Figures 25 and 26. The mean thrust on the blades is predicted very well with pseudo motion, staying within 5% of the real motion results. The thrust fluctuation, however, varies substantially, in particular for sway cases. The sway case behaves
differently from the others because as the rotor moves to the side, the undisturbed flow is allowed to energize the wake, moving it further away from the rotor in the axial direction. Concurrently, the rotor moving to the side means it is moving away from the wake induction in the lateral direction, or moving outside of the stream tube and into the freestream flow. Thus a rotor in sway achieves lower axial induction and higher thrust, which is a counter intuitive combination. The variation in $\Delta C_{Tb}$ for sway in pseudo and real motion can also be explained by this interaction of the freestream flow and the wake. If we consider that on
one side of the rotor the wake is stretched by the aforementioned interaction and on the other side the wake is compressed by the opposite effect, $\Delta C_{Tb}$ can be dampened or augmented by the real sway motion, depending on how the blade position is aligned with the lateral motion.

Figures 27, 28, and 29 show the wakes for real and pseudo surge, sway, and yaw, respectively. The surge case uses axial symmetry, hence only one blade is shown. The differences between real and pseudo motion in surge are subtle, while the yaw
motion is more obvious, as the wake becomes more irregular in real motion, due to the lack of symmetry. The sway case is quite extreme, with the pseudo motion showing a very well behaved wake, almost identical to the other cases, while real motion makes the wake become chaotic very quickly.

The amplitudes and frequencies used in this paper for surge are mostly related to the UNAFLOW experiment. For sway and yaw, we decided to use the same frequencies as in surge, adapting the amplitude to obtain values of $\Delta C_{Tb}$ in the same order of
330 magnitude as the surge cases. While this is an arbitrary decision, we believe the results herein can be generalized. For surge, as long as the amplitude and frequencies do not lead to the wakes impinging on the blades, we have no reasons to believe the

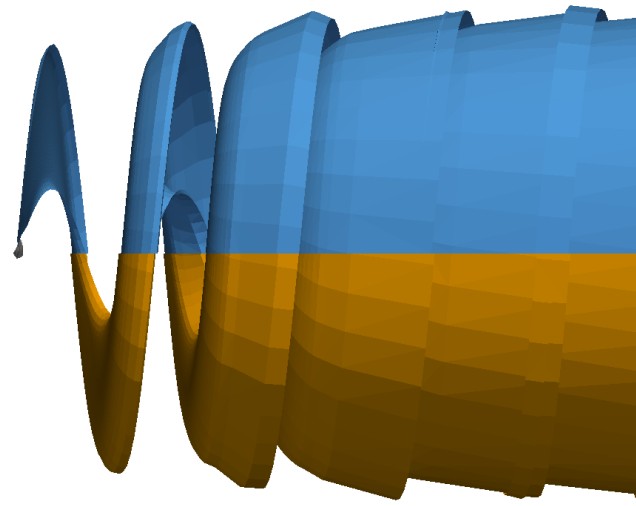

**Figure 27.** Wake for real (top, blue) and pseudo (bottom, orange) surge motion at $f = 1$ Hz. Only one blade shown.

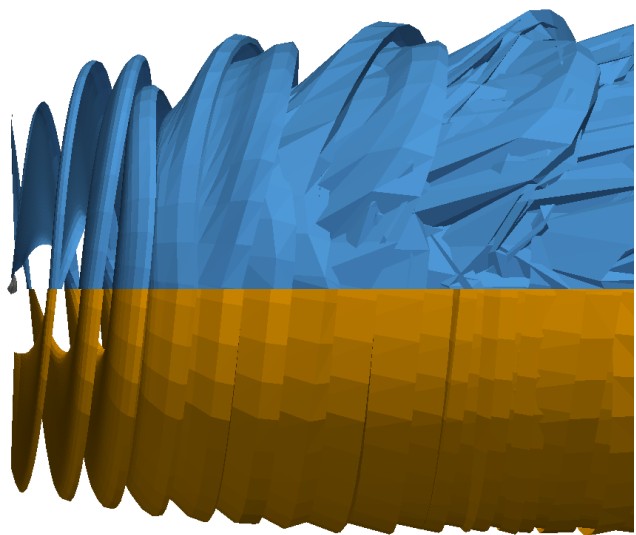

**Figure 28.** Wake for real (top, blue) and pseudo (bottom, orange) sway motion at $f = 4$ Hz.

effect of wake motion will be severe. For sway, the amplitude chosen for this section is compatible with the turbine oscillating from side to side by less than $4°$, which seems realistic for a FOWT. This is sufficient for the thrust fluctuations to be severely affected by wake motion, leading to the conclusion that sway motion is heavily affected by wake motion effects. Finally, for dynamic yaw motion, with very high yaw amplitudes, a similar effect of wake motion as seen on sway could be encountered.

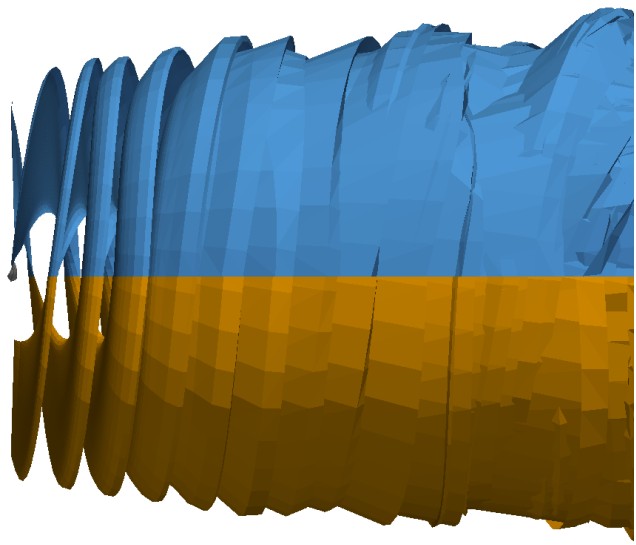

**Figure 29.** Wake for real (top, blue) and pseudo (bottom, orange) yaw motion at $f = 3$ Hz.

However, we believe FOWT platforms do not tend to yaw dynamically at high amplitudes. Hence, we expect the wake motion effects for yaw to be small, in general.

## 6 Conclusions

We have shown that a free wake panel method can accurately capture mean and unsteady thrust of a surging wind turbine. The
340 methodology used in this paper slightly underpredicts the mean thrust and overpredicts the amplitude of thrust fluctuations, however results are comparable and in line with the state-of-the-art (Mancini et al., 2020).

The effects of the rotor motion on the tip vortices was also shown to be accurately captured by the method in what we believe is the first simulation of surging wind turbine wakes that accurately reproduce experimental data. Wake vortices are particularly difficult to capture with CFD methods, as the Eulerian approach tends to dissipate them (Bayati et al., 2018a).
Lagrangian methods have a significant advantage in preserving the wake vortices near the body, with the disadvantage of wake entanglement far from the rotor, which in turn requires some dissipation for stabilization.

We found that the surge frequency had to be tripled from its maximum value in the experimental campaign to reach a nonlinear response in thrust. The current method allowed us to investigate this by isolating Theodorsen effects. This means that, in reality, the nonlinear response could happen earlier due to other phenomena, such as dynamic stall.
We then studied side wind, sway motion, and yaw motion of a rotor. We demonstrated the complexity of the forces acting on the blades during sway and yaw motions, even without flow separations. By comparing side wind results with sway results using the maximum sway angle and including the sway velocity in the thrust coefficient, we were able to show linear behavior for sway at low frequencies that matched the side wind trends. For the yaw motion, the blade tip surge effect was demonstrated

by investigating the axial force on a single blade during a yaw cycle. For both sway and yaw, the blade forces fluctuations can vary by a factor of 2, depending on the ratio between the rotor motion and rotation, and the blade initial position.

Finally, we used an interesting feature of the current methodology to perform what we refer to as pseudo motion simulations, where we accounted for the surge, sway, and yaw motion on the rotor, without actually performing these motions on the turbine. With this we showed the sensitivity of wake deformation on the forces on the blades. It was found that the sway motion allows undisturbed air to enter the wake, increasing the mean thrust and, in our case, reducing the dynamic loads. Surge and yaw were shown to be fairly insensitive to the wake motion, which explains the fact that methods that do not capture wake dynamics can still predict surge motion effects well.

This work is a stepping stone towards building a tool that is able to simulate FOWT in a way that is accurate, robust, and efficient. Future work will expand to more complex cases, such as full offshore platform motion and aeroelasticity.

*Data availability.* The data generated in this work can be provided by request to the first author.

*Author contributions.* André Ribeiro developed the panel code, performed the simulations, post-processed the data and wrote the paper. Damiano Casalino contributed to the conceptualization of the study and in the interpretation of some results. Carlos Ferreira contributed to the conceptualization of the study, interpretation of the results, and the development of the panel code.

*Competing interests.* There are no competing interests.

*Acknowledgements.* The authors are grateful to Felipe Miranda for providing the experimental data (Fontanella et al., 2021b). We are also very grateful for the help of Massimo Gennaretti and Riccardo Giansante in achieving accurate results for dynamic motion in the panel code.

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
