# Peer review of "Nonlinear Inviscid Aerodynamics of a Wind Turbine Rotor in Surge, Sway, and Yaw Motions Using a Free Wake Panel Method"

_Wind Energy Science, 2023_

## Author Response (AR1)

Reviewer 1:

Thank you for the thorough review and valuable feedback. We made changes to the original manuscript, based on your comments, and we believe the paper benefitted from these changes.

Here are answers to your questions and comments:

Q: For the investigation of sway motion the authors start with what they call side wind conditions as their static reference case. They perform these simulations to a maximum side wind angle β of 10° even though the reach up to 40° in their sway motion. Why did the authors only cover such a small region in the side wind simulation and not the expected whole range if they want to compare the results to the sway motion?

A: Thank you for the question. We extended the side wind runs for better comparison (up to 40deg, showing nonlinear range).

Q: The results in figure 12 show the time history of the thrust coefficient of a single blade for different reduced frequencies. The plots for all reduced frequencies show an oscillation on larger scales than the rotational frequency or the sway frequency. Could the authors comment on that?

A: Yes. This is discussed in Fig. 16 and is due to the interaction between the sway motion and the rotation. Fig. 17 (time history of horizontal sway velocity projected on airfoil chord at the tip of blade 1) was added to make the explanations clearer. I also added Equations 3 and 5, which are for the horizontal sway velocity projected on airfoil chord and streamwise blade tip yaw velocity, respectively.

Q: In figure 14 it would help if the authors could add a sketch of the rotor position and the sway motion for each angular position shown in the figure. Also, there seems to be a phase shift between the rotor position and the sway position. Why do the author not discuss and show this like they do for the surge motion?

A: Thank you for the suggestion. I added Fig. 18 with the requested sketch and Figs. 17 (explained above) and 22 (time history blade 1 streamwise blade tip yaw velocity), which I believe clarify the explanations for sway and yaw. I also updated Fig. 21 (previously 17), in favor of a more concise explanation for yaw. The azimuth and sway motion are aligned. The phase shift is between the force and the azimuth/sway, which we now comment on. Performing a quantitative comparison of the phase encountered in sway and yaw motion is difficult. As seen in Fig 14, some of the peaks and troughs seem to align with the azimuth, while others do not. Taking a Fourier transform of the

signal leads to multiple peaks, not only at the sway and rotation frequencies. Hence, there are many variables involved and we could not think of an elegant way to create something as simple as Fig 10 for sway and yaw.

Q: Even though it is interesting to see what happens to the single blade thrust it would be also very interesting to see how that effects the total thrust of the turbine. Does the total thrust also show oscillations? Additionally, the results for a single blade depend on the starting condition where here the blade is in the 12 o'clock position when the sway motion or at its maximum. Does the total thrust also depend on the starting conditions?

A: As alluded to in the original text, the total thrust is fairly stable, as there is a cancellation effect with the 3 blades. We added text in the second paragraph of Section 4.1 to clarify this.

Q: The authors mention, that they simulated all three rotor blades for the sway and yaw experiments. So they do have the data for different starting positions at hand. Why don't they show these and discuss the results and maybe the differences? Why do the authors think that the starting position at 12 o'clock is representative for all the other cases? Maybe it is, but it is not clear to reader.

A: Thank you for the question. We added Section 4.4 (Effect of Blade Azimuth) to clarify the effects of blade choice, showing the possible range of thrust fluctuation for arbitrary combinations of frequencies and blade positions. We also updated Figures 15 and 20 to show all 3 blades. The aforementioned Equations 3 and 5 should also help clarify this.

Q: What is the purpose of figure 15? It is impossible to actually see any differences in the different plots and the only remark is that is very similar to the behaviour in figure 12.

A: It is indeed quite similar to Fig. 12 and it serves mostly a qualitative purpose of indicating that the behavior is similar to sway, but more complex. This can be observed by the symmetry seen in the sway, but not in most yaw time histories.

Q: Figure 17 would also benefit from additional sketches of the rotor position like proposed for figure 14.

A: Thank you for the suggestion. Fig. 18 was added as a reference for these analyses. Also, Fig. 22 (mentioned above) was added to make explanations even clearer.

Q: I do not really understand why the authors use different methods to analyse the data from the three cases. In the surge case they look at the impact of the reduced frequency on the total thrust where they later only look at one blade and their individual thrust with respect to side wind angles and tip yaw velocities. This could be motivated in more detail.

A: The reason for focusing on a single blade is due to the cancellation effects seen in the rotor forces. This was clarified in the text. The full rotor thrust fluctuation is quite small (by over an order of magnitude) compared to the single blade thrust fluctuation. The reason to use beta is to compare with static side wind. The reason to use the surge velocity is to compare surge and yaw, even with different amplitudes. This was clarified in the text.

Q: Section 5 seems to be a little out of place. I do not see the connection to the rest of the paper and the results presented seem to be more or less randomly and the authors state that the results for other cases might be different or even worse. This topic seems to be a paper on itself where the authors should discuss the results in more detail and try to work out more qualitative results than just the pictures from the wakes.

A: Thank you for the comment. I'm not sure this study could be a full paper in itself. We believe Section 5 is linked to the findings of the rest of the paper, first when it comes to surge, where other publications have focused on. Methods that do not take wake deformation into account seem to achieve good results and Section 5 addresses that. Second, it justifies the nonlinear findings in sway. We linked the new Section 4.4 to Section 5 by noting the behavior of the sway forces not following the simple assumptions of Equation 3. Hence, an explanation for what is happening during sway, with the rotor moving out of the stream tube, is necessary. Although the values used in the paper are arbitrary, they should be representative and we made changes to the last paragraph of Section 5 to reflect that.

Q: line 248: „"A more irregular pattern than that of sway is observed". This should be surge, since that is mentioned in the sentence before and is shown in the corrersponding figure.

A: Thank you for the suggestion. It was indeed a comparison with sway, but the text was not clear before. I have tried to make it clearer now

Thank you for the review and valuable feedback. We made changes to the original manuscript, based on your comments, and we believe the paper benefitted from these changes.

Here are answers to your questions and comments:

Q: The title seems to generic given that the aims of the paper are quite specific. It is recommended to change the title to reflect the contents of the work more specifically.

A: Thank you for the suggestion. I changed the title to something more specific: "Nonlinear Inviscid Aerodynamics of a Wind Turbine Rotor in Surge, Sway, and Yaw Motions Using a Free Wake Panel Method".

Q: The paper would benefit from a nomenclature section.

A: Thank you for the suggestion. I added a nomenclature section.

Q: The abstract and introduction are well written in general but the research questions and objectives found in the introduction do not highlight clearly the work shown in section 5. This must be corrected.

A: Thank you for the suggestion. We added a paragraph explaining the motivation behind understanding wake motion and its importance in improving lower fidelity modes.

Q: There are large chunks of text that are the same as the conference article mentioned by the authors. Perhaps it would have been ideal to make reference to the previous articles and replace these identical snippets of text with shorter descriptions of the methodology and put more focus on those aspects of the methodology that are more pertinent to this study (see specific comments).

A: Thank you for the suggestion. We removed some unnecessary text (an entire paragraph from Section 3 and 3.2, two from 3.3).

Q: Section 2 Methodology – Some more detail should be provided on how the model handles the side panels on the blade (for example the side of the tip) since this has crucial implications on the sway aerodynamics. For instance, at those blade angles where the flow is normal to the blade edge face, is any vorticity released in the flow direction along the blade span? This would be better than simply repeating the text from the previous paper. Having said this, the points mentioned in the methodology are still important to at least mention.

A: Thank you for the interesting question. I updated Figure 3 to show the blade tip panels and added this clarification: "The blade tips and roots are closed to enforce impermeability. Wake panels are only shed from the trailing edges, meaning no vorticity can be shed from the 90deg edges at the tips and roots."

Q: Figures 7 and 8 – It is somewhat difficult to compare these results side-by-side. It is recommended to overlap these results on a single figure with the vortex filaments having a distinct colour compared to the colour palette used with the experimental results. An x and y-axis scale would also help to underline the differences in vortex filament convection.

A: Thank you for the suggestion. Adding both on top of each other was overwhelming, so instead I added the pink dots that indicate the numerical tip vortices in Fig 8 to Fig 7 as well. That way a direct comparison can be made.

Q: Pg 8 line 151-152 – "The simulations shown here employed the aforementioned vortex core model and achieved better wake stability with it.". can the authors be more specific on what they mean by better wake stability (is it referring to numerical stability).

A: We meant stability in the sense of wake entanglement. I clarified this in the text.

Q: Section 4.1 – To aid the explanation could the authors indicate the angle beta on a diagram and distinguish this from the dynamic yaw angle. This can also be done in the methodology section.

A: There is no real difference between beta and the yaw angle, other than the yaw angle changing over time. In any case, I created diagrams for beta and beta_max (new Figures 11 and 13).

Q: Figure 13 – Can you explain in a bit more detail the drop of delta Ctb that is observed at beta approximately 37degrees? Figure 14 - Can you explain in a bit more detail the drop of delta Ctb that is observed at Usy/U\infty approximately 0.4?

A: I added equations (Eq. 3 and 5) that approximate the sway and yaw Ctb curves and a new section that explains the behavior of the thrust and its link to blade azimuth and ratio between frequencies (sway or yaw ratio with rotation frequency). I believe this clarifies the behavior.

Q: Section 5 – This discussion seems to address a different research question than originally posed in the introduction. It is recommended to expand more on this feature of the paper in the introduction. Another option is to remove this discussion entirely.

A: Thank you for the suggestion. Indeed, the motivation behind this was not mentioned in the introduction. I added a paragraph, explaining the potential importance of understanding wake motion for low fidelity methods, especially when it comes to sway and yaw. I also linked this discussion to the end of Section 4.